# Prevalence and Genotyping of HPV in Oral Squamous Cell Carcinoma in Northern Brazil

**DOI:** 10.3390/pathogens11101106

**Published:** 2022-09-27

**Authors:** Silvio Augusto Fernandes de Menezes, Yasmim Marçal Soares Miranda, Yngrid Monteiro da Silva, Tábata Resque Beckmann Carvalho, Flávia Rayane Souza Alves, Rodrigo Vellasco Duarte Silvestre, Aldemir Branco Oliveira-Filho, Tatiany Oliveira de Alencar Menezes, Ricardo Roberto de Souza Fonseca, Rogério Valois Laurentino, Luiz Fernando Almeida Machado

**Affiliations:** 1School of Dentistry, University Center of State of Pará, Belém 66060-575, PA, Brazil; 2Evandro Chagas Institute, Health Ministry of Brazil, Ananindeua 67030-000, PA, Brazil; 3Study and Research Group on Vulnerable Populations, Institute for Coastal Studies, Federal University of Pará, Bragança 68600-000, PA, Brazil; 4School of Dentistry, Federal University of Pará, Belém 66075-110, PA, Brazil; 5Biology of Infectious and Parasitic Agents Post-Graduate Program, Federal University of Pará, Belém 66075-110, PA, Brazil; 6Virology Laboratory, Institute of Biological Sciences, Federal University of Pará, Belém 66075-110, PA, Brazil

**Keywords:** squamous cell carcinoma, oropharyngeal cancer, HPV, viruses, infectious diseases

## Abstract

Highly oncogenic human papillomavirus (HPV) is well known to be associated with and a risk factor for various types of oral carcinomas such as oral squamous cell carcinoma (OSCC). The aim of this study was to evaluate and describe the HPV-induced OSCC prevalence and genotyping in the city of Belém, northern Brazil. This cross-sectional study features 101 participants who attended an oral pathology referral center in a dental college looking for diagnoses of oral lesions (OL). After signing the consent term and meeting the inclusion criteria, all participants went through a sociodemographic and epidemiological questionnaire. Then, OL were collected by excisional or incisional biopsy depending on OL size; after that, OL tissues were preserved in paraffin blocks to histopathological diagnoses. Afterwards, paraffin blocks were divided into benign and malignant/premalignant lesions based on the classification of potentially malignant disorders of the oral and oropharyngeal mucosa. Then, the paraffin blocks had DNA extraction performed by the ReliaPrep FFPE gDNA Miniprep method in order to identify HPV DNA of high oncogenic risk and low oncogenic risk. Then, the viral DNA was amplified and typed using the Inno-Lipa genotyping Extra II method, and the collected data were analyzed by Chi-square and G-tests. In total, 59/101 (58.4%) OL were malignant/premalignant lesions, of which OSCC was the most prevalent with 40/59 (67.7%) and 42/101 (41.6%) benign lesions. The most common area of OL incidence was upper gingiva 46/101 (45.5%). Regarding HPV DNA detection, approximately 27/101 (26.7%) had positive results; of these, 17/59 (28.8%) were malignant/premalignant lesions, and the most prevalent genotypes detected were 16, 18, 52 and 58, while among benign lesions, 10/42 (66.6%) had HPV-positive results, and the most prevalent genotypes detected were 6, 11 and 42. Age range was the only risk factor with a significant association between HPV and OSCC presence (*p*-value: 0.0004). A correlation between OSCC and oral HPV among analyzed samples could not be demonstrated in our small cohort.

## 1. Introduction

The human papillomavirus (HPV) is a member of the Papillomaviridae family. According to Monteiro et al. [1], there are more than 130 species and nearly 228 HPV genotypes have been identified so far—all with the capacity to tropism for mucosal and cutaneous epithelia, such as squamous tissue. As explained by the International Committee on Taxonomy of Viruses (ICTV), HPV is a small non-enveloped double-stranded circular DNA virus approximately 52–55 nm of diameter, which is composed by a protein icosahedral capsid made of 72 pentameric capsomeres that surround the viral genome with 8000 nucleotide base pairs [2,3].

HPV can infect the epithelial surface through microlesions at the basal layer, during unprotected sexual behavior, which then conduct to a proliferative benign or malignant lesion both in skin, oropharyngeal mucosa and the vaginal or anal tract due to morphological similarities [4]. Currently, HPV is designated into two forms: high oncogenic risk (HR) and low oncogenic risk (LR) subtypes. LR subtypes 6, 11, 40, 42, 43, 44, 54, 61 and 70 are associated with benign lesions such as condylomatous warts; HR subtypes include HPV 16, 18, 26, 31, 33, 35, 39, 45, 51, 52, 53, 56, 59, 66, 68, 72, and 81 that are linked to malignant lesions such as squamous cell carcinoma (SCC) [5,6].

HPV oncogenic potential is based on the virus capacity to lodge two specific encode viral oncoproteins, E6 and E7 genes, into hosts, infecting the cells’ genome [4,7,8]. This capability of HPV promotes the invalidation of the activities of important tumor suppressor and apoptosis proteins, such as Tumor Protein p53 (TP53) and Retinoblastoma Protein (pRb). E6 viral oncoproteins interact with TP53, causing its degradation due to its relationship with E6-Associated Protein [9,10,11]. Regarding E7 viral oncoproteins, it attaches to pRb, inactivating its capacity to inhibit excessive cell cycle progress and therefore resulting in carcinogenic activity in the oropharyngeal and genital epithelium [9,10,11].

Worldwide, HPV infection is a major public health issue, mainly because it is one of the most common sexually transmitted infections (STIs). Globally, HPV infection prevalence is approximately 12% with continental discrepancies due to the socioeconomic development and vaccination program of each country, and HPV-induced cancer has a prevalence of 5.1%, which ranges through genders and innumerous anatomic sites [12,13]. In Brazil, according to Colpani et al. [14,15], HPV has a national prevalence of 25.41% with some anatomic site variation as: penile region, 36.21%; anal region, 25.68%, oropharyngeal, 11.89% and among this prevalence, 17.65% was associated to HR HPV subtypes.

SCC is one of the main malignant lesions of invasive skin cancer and is easily identified by an atypical, accelerated increase in squamous cells. Following the morphological similarities with oropharyngeal mucosa tissue, the presence of an oral SCC (OSCC) is possible; thus, OSCC can arise from any location of oropharyngeal mucosa [16,17]. According to van der Waal [18] and Jiang et al. [19], the most frequently affected sites are the tongue mouth floor, sublingual area, gingiva, hard palate and lips. As claimed by Syrjänen [20], Tumban [21] and Panarese et al. [22] OSCC, clinically, it appears as an ulcerative lesion with a necrotizing central area and lifted borders.

In general, for OSCC, well-established risk factors are smoking, alcohol overconsumption and tobacco chewing, although since 1983, it was hypothesized that OSCC can emerge from HR HPV subtypes; in particular, HPV subtype 16 (HPV-16) is associated with unprotected sexual behavior. However, the specific role of HR HPV regarding OSCC risk factors is not fully understood. Therefore, furthermore information is needed to clarify the relationship between HR HPV infection and OSCC in northern Brazil. So, in this context, this study aimed to evaluate and describe the HPV-induced OSCC prevalence and genotyping in the city of Belém, northern Brazil.

## 2. Materials and Methods

This descriptive, cross-sectional single-center study was population-based on clinical symptoms, sociodemographic and epidemiological data from individuals who attended an oral pathology and malignant lesions referral center at a dental college (CESUPA) located in the city of Belém, Pará, northern Brazil (Figure 1). All individuals who attended this referral center during the period from January 2019 to December 2019 were invited to participate in the study, and of these, 101 individuals met the inclusion criteria and were diagnosed with oral benign and malignant/premalignant lesions/tumors.

All interventions were performed in accordance with the guidelines and regulatory standards for research involving human subjects of the National Health Council and Papilloma Virus laboratory of Instituto Evandro Chagas (IEC). This study was approved by the Ethics Committee on Human Research of the University Center of the State of Pará—CESUPA under protocol number 4.197.815. Written informed consent was obtained from all 101 patients for the publication of any potentially identifiable images or data included in this paper.

### 2.1. Clinical Parameters

The benign and malignant lesions were established according to the classification of potentially malignant disorders of the oral and oropharyngeal mucosa [18]. OL were sub-divided in situ into benign lesions—traumatic fibroma, focal epithelial hyperplasia, pyogenic granuloma, papilloma, verruca vulgaris, condyloma acuminatum—and malignant or premalignant lesions: oral squamous cell carcinoma, leukoplakia, erythroplakia, oral lichen planus, oral submucous fibrosis and carcinoma. Although the main objective of the study is to associate HPV infection and SCC, both OL were biopsied, and benign lesions were used as a control group to evaluate HPV prevalence in different lesions [23,24].

### 2.2. Sample Collection and Processing

The sample consisted of patients registered and treated at CESUPA. In total, 101 individuals were informed about the purpose of the study and invited to participate. Then, they all agreed to and signed a written consent form before data collection and oral evaluation. The study eligibility criteria were: (i) ≥18 years old; (ii) have a conclusive diagnosis of oral benign lesion or OSCC; (iii) resident of Pará State; (iv) medical records filled in; and (v) signed the free and informed consent form. The exclusion criteria were: (i) individuals who transferred to other cities, affecting follow up; (ii) individuals with neurological and/or cognitive impairment; (iii) medical records not filled out correctly; and (iv) refusal to sign the consent form. Individuals who met the inclusion criteria were invited to participate in this study and signed the consent form.

Each participant was orally evaluated in a private location in the oral pathology department. Clinical data were collected by a single researcher, a specialist in oral pathology who had previous experience in clinical studies. The intraoral clinical examination was performed in a dental office, in a dental chair, under indirect and artificial light, using a dental mirror and clinical tweezers, which were all sterile, consisting of disposable materials; the OL evaluations were performed daily.

Demographic and epidemiological data were obtained through a pre-tested standardized semi-structured questionnaire and medical records. Regarding oral biopsies, all lesions included had been submitted either to excisional biopsy, when OL were sized approximately to ≤1 cm, or incisional biopsy, when they were sized approximately to size ≥ 1 cm with a scalpel. All biopsies were performed by only one researcher who had previously experience in clinical biopsy. Examinations occurred under local anesthesia, and for excisional biopsies, a 3 mm margin of normal tissue was included. Then, the biopsy material was preserved in 10% formaldehyde then transported to histopathologic exam.

All pieces were processed in paraffin blocks for histological analysis. At first, it was necessary to replace tissue liquid with paraffin; then, a sequence of ethyl alcohol baths at increasing concentrations (70–99%) for approximately 6 h for tissue dehydration was made. Subsequently, the pieces went through the process of diaphanization in xylol for 3 h and impregnation in molten paraffin at 60 °C for 2 h. Finally, the pieces were transferred to steel molds, which were bathed in liquid paraffin at 65 °C and taken to rapid cooling at 0 °C [25]. After preparation, two different pathologists evaluated the hematoxylin and eosin-stained sections of all lesions for confirmation of the diagnosis. A microscopic diagnosis was rendered according to the WHO classification of potentially malignant disorders of the oral and oropharyngeal mucosa [18].

### 2.3. DNA Extraction from Paraffin Samples

Each block of paraffin samples was cut into 10 “slices” with 5 µm thickness each, and viral DNA extraction was performed using the “ReliaPrep FFPE gDNA Miniprep Syste (Promega Corporation, Madison, USA). This system is based on whether in the use of cellulose membranes in columns, where the lysed biological material is subjected to centrifugation, the DNA is bound to membranes charged with (+) charge so that there is binding to the (−) DNA. Subsequently, washes were carried out with alcoholic solutions, and the DNA elution was carried out in a saline medium. All conditions described were specified in the manufacturer’s protocol.

Each “sample” was heated in a thermoblock at 80 °C for approximately 2 min, and 500 μL of mineral oil was added to dissolve the paraffin—a process which was repeated for all “slices”. Subsequently, 300 μL of PBS buffer and 20 μL of Proteinase K were added for sample digestion. Soon, the samples were incubated at 65 °C for 1 h. After the complete digestion process, the DNA samples were considered homogeneous; then, they were transferred to another Eppendorf tube and were incubated at 95 °C for 15 min. After this process, the samples were kept at room temperature, and later, 220 μL of lysis buffer and 240 μL of absolute ethanol were added to assist in the aggregation of precipitated DNA. All samples were transferred to kit columns and then centrifuged at 10,000 RCF for 3 min. The fluid remaining in the tube was discarded, and 500 μL of washing solution was added, centrifuging at 10,000 RCF for 30 s with the cap closed and 16,000 RCF for 3 min with the tube cap open. Subsequently, the column was transferred to another Eppendorf tube, discarding the collection tube. Finally, 50 μL of elution buffer was added directly to the column, and it was centrifuged again at 16,000 RCF for 1 min. The column was discarded and the Eppendorf tube with the filtered sample was stored at −70 °C.

### 2.4. HPV Detection and Typification

The extracted viral DNA was identified and typed in the analyzed samples using the “Inno-Lipa Genotyping Extra II System (Fujirebio, Tokyo, Japan), which is able to amplify a portion of the HPV genome in the viral L1 gene region. Through the reverse hybridization system, this generated fragment is able to identify infection by up to 32 viral types. Therefore, the presence of up to 28 genotypes of HPV (high oncogenic risk: 16, 18, 26, 31, 33, 35, 39, 45, 51, 52, 53, 56, 58, 59, 66, 68, 69, 73, 82; and low oncogenic risk: 6, 11, 40, 42, 43, 44, 54, 61, 70) according to the manufacturer’s instructions.

This system is based on the hybridization of the PCR-amplified fragment through the base homology of the amplified fragment with the probe contained in a “nylon” strip that corresponds to the specific sequence for each of the mentioned types, which defines, in addition to positivity, the infecting viral type. So, the viruses were cataloged on a decreasing scale as to their oncogenic risk.

### 2.5. Statistical Analysis

The collected data were analyzed by the BioEstat program and evaluated with respect to mean, standard deviation and absolute and relative frequency, as well as *p* value (*p* < 0.005) by the Chi-square, Fisher Exact Test and G Test, in the selected groups.

## 3. Results

### 3.1. Sample Characteristics and Anatomical Subsites

In total, 101 individuals were recruited for study assessment, and the sociodemographic and behavioral data are presented in Table 1: 75/101 (74.2%) in the Belém metropolitan area and 26/101 (25.8%) in the countryside area; of those data, malignant lesions (59/101, 58.4%) were more prevalent than benign lesions (42/101, 41.6%). In the metropolitan area, Belém city was the most visited site to diagnose and treat the lesions. Belém presented 36/75 (48%); of these, 20/59 (33.8%) were malignant lesions and 16/42 (38%) were benign lesions. Ananindeua was the second city with 20/75 (26.6%); of these, 13/59 (22%) were malignant lesions and 7/42 (16.6%) were benign lesions. The third most popular area was Santa Bárbara with 10/75 (13.3%); of these, 6/59 (22%) were malignant lesions and 4/42 (16.6%) were benign lesions. In the countryside area, Santa Izabel was the city with most biopsies, 9/26 (34.5%); of these, 5/59 (8.4%) were malignant lesions and 4/42 (9.5%) were benign lesions. Next was Abaetetuba with 6/26 (23%); of these, 4/59 (6.7%) were malignant lesions and 2/42 (4.7%) were benign lesions. Barcarena had 4/26 (15.3%); of these, 3/59 (5.6%) were malignant lesions and 1/42 (2.4%) were benign.

The average age was 39.9 years and had a significant *p* value of 0.0004 (*p* < 0.005); the 60–69 age range was the most prevalent among individuals with 25/101 (24.7%). Of these, 19/59 (32.2%) were malignant lesions and 6/42 (14.2%) were benign lesions. Thus, we may infer, in our sample, that age can influence OSCC prevalence. Otherwise, most of the samples were: female 55/101 (54.4%)—of these, 29/59 (49.2%) were malignant lesions and 26/42 (62%) were benign lesions; mixed ethnicity 46/101 (45.5%)—of these, 26/59 (44%) were malignant lesions and 20/42 (47.6%) were benign lesions; smoking habits 71/101 (70.3%)—of these, 37/59 (62.7%) were malignant lesions and 34/42 (81%) were benign lesions; and alcohol overconsumption 71/101 (70.3%)—of these, 44/59 (74.5%) were malignant lesions and 27/42 (64.3%) were benign lesions. In addition to age range, no other significant association between risk factors, sociodemographic factors and type of lesion was found.

In Table 2, the anatomical subsites were evaluated searching for significant associations between malignant lesions and anatomical subsites. No significant association was possible (*p* < 0.4713), although upper gingiva was the most common area of incidence (46/101, 45.5%) of both malignant lesions (24/59, 40.6%) and benign lesions (22/42, 52.3%).

### 3.2. Histological Analysis

In Table 3, the histopathological exams results of both benign and malignant lesions were presented. In malignant lesions, the most prevalent oral manifestation was OSCC 40/59 (67.7%), leukoplakia 9/59 (15.2%) and erythroplakia 7/59 (11.8%). In benign lesions, the most prevalent was traumatic fibroma 20/42 (47.6%), condyloma acuminatum 8/42 (19%) and focal epithelial hyperplasia 5/42 (12%), and a significant *p* value of 0.0001 (*p* < 0.005) was established.

### 3.3. HPV Prevalence and Typification

In total (*n* = 101), 27 (26.7%) had positive results for HPV DNA and 74 (73.3%) had negative results for HPV DNA (Table 4). Among samples with malignant/pre-malignant lesions, 28.8% tested positive for HPV. In this sample with HPV (*n* = 17), several cases of LO were detected. OSCC (*n* = 12; 70.5%) predominated, but cases of leukoplakia (*n* = 4; 23.5%) and erythroplakia (*n* = 1; 6.0%) were also recorded. HPV genotypes 16, 18, 52 and 58 were detected, and all were classified as high oncogenic risk. In the ten samples with HPV-16, OSCC (*n* = 7; 70%), leukoplakia (*n* = 2; 20%) and erythroplakia (*n* = 1; 10%) were detected. OSCC was also detected in five (29.4%) samples of HPV-18, and leukoplakia was recorded in each of the HPV-52 (5.9%) and HPV-58 (5.9%) positive samples.

Among the benign lesions, 23.8% tested positive for HPV. Genotypes 6, 11 and 42 were detected, and all were classified as low oncogenic risk. Verruca vulgaris (50%), papilloma (33.3%) and condyloma acuminatum (16.7%) were detected in samples with HPV-16. Condyloma acuminatum was detected in all HPV-11 samples, and papilloma was detected in the HPV-42 sample.

## 4. Discussion

Since 1983, the increasing number of papers indicating a correlation between HPV infection with oropharyngeal tumors or lesions has increased [4,5,6,9,11,17,18,19,20,21,22,23,24,26,27,28,29,30]. The present study evaluated the prevalence of HR HPV using fresh and frozen biopsied samples; through various oral lesions (OL), OSCC was included and associated with patients’ sociodemographic parameters in northern Brazil. To the best of authors’ knowledge, this is the first epidemiological cohort of this type in northern Brazil. However, in this study, it was not possible to demonstrate a direct correlation between HR HPV and OSCC among the analyzed samples, even with different methods regarding comparisons with other studies such as analysis of freshly frozen OSCC samples to improve positive results in histopathological analysis (the same method as Drop et al. [31]) and having a benign lesion control group.

Through the years, OSCC-related HR HPV has been well documented, although different important factors could affect the outcomes of HPV-induced OSCC. The most common associations are smoking and alcohol overconsumption, which according to the literature influences various populations worldwide. Madathil et al. [32] evaluated in Montreal, Canada, 631 participants with smoking habits and its relationship with HPV-associated oropharyngeal tumors; of these, 40% were HPV positive and smokers, showing a higher prevalence compared to the control group (16%), and HPV-16 was the most prevalent subtype among participants. Auguste et al. [33] investigated the influence of tobacco and alcohol joint consumption with HPV and the occurrence of head and neck SCC among 550 individuals in Guadeloupe and Martinique, French Caribbean. The authors demonstrated that the combination of tobacco and alcohol consumption could induce a symbiotic effect on the incidence of OSCC, and HPV-52 was the most prevalent subtype.

Smith et al. [34] evaluated 201 participants with smoking and alcohol consumption habits, HPV presence and its relationship with head and neck SCC in Iowa, United States. The authors demonstrated a prevalence of 46% HPV-positive cases associated with smoking and alcohol consumption, and HPV-16 was the most prevalent subtype among participants. In Brazil, Rodrigues et al. [30] evaluated the prevalence of oral HPV in various OL among 278 people who use crack-cocaine (PWUCC) in the cities of Bragança and Capanema in northern Brazil; of these, 111 (39.9%) PWUCC were HPV positive, and HPV-16 was the most prevalent subtype among PWUCC. Another interesting fact corroborated by various studies is that the increased smoking and alcohol consumption associated with the presence of HR HPV will influence in the severity of OL.

In this study, risk factors such as smoking (70.3%) and alcohol overconsumption (70.3%) proved to be an important etiology factor to OSCC, confirming the results of all the studies mentioned above, although among our samples, the smoking and alcohol overconsumption associated with HR HPV prevalence could not be inferred, which was similar to the other studies prior. Perhaps the influence and the prevalence of HR HPV over OL might be impacted by other risk factors besides smoking and alcohol overconsumption, such as unsafe sex, multiple sexual partners, drug dependence, lack of access to public health services, co-infections such as HIV-HPV, and HPV non-vaccination, which would facilitate HPV throughout the state [1,25].

Bezerra et al. [35] demonstrated that smoking and alcohol overconsumption, associated or individually, are important risk factors to a higher prevalence of OSCC. The authors also stated that tobacco and ethanol may increase oral epithelium permeability by the exposure of oral tissue to various carcinogenic agents presented in tobacco and ethanol. It can also decrease interleukins (IL) expression, such as IL-18 and DDX3 protein, which regulate the cell cycle and control the progression of malignant neoplasms, as well as increase COX-2 pro-inflammatory activity. In our study, one of the main risk factors was age (*p* < 0.05). According to our results, age range (60–69) could be considered as a major influencing factor for high OSCC prevalence; however, a cause–effect relationship could not be established.

In the present study, the OSCC prevalence was 58.5% (59/101) and the HPV prevalence was 27/101 (26.7%). Among HPV positive samples, the most prevalent HR HPV subtype was HPV-16 (10/17—58.8%), which was exactly the same subtype as previous studies. When comparing the HPV prevalence ratings to our studies, ours is the smallest prevalence, which is mainly because our cohort had a smaller sample size. Other influencing risk factors might be demographic region, socioeconomic status, anatomical sites and access to quality public health services. Bean et al. [36] demonstrated that populations with a lower income, living in countryside areas or overly dense urban areas are more likely to present severe cases of late-stage cancer. Our smaller HPV prevalence could be due to Brazil’s northern region having many cities surrounded by hydrographic access routes, which make it difficult to access health care. In addition, the distances between major urban areas and countryside cities are higher than in southern Brazil [37,38].

Despite the fact that OSCC prevalence was the main objective of our study, during the results analysis, the decreased prevalence of HPV was interesting, because it is different from the literature. The higher prevalence of HR HPV subtypes among HPV prevalence highlighted the urgency to improve the distribution of specialized oral pathology services, HPV vaccination and implementation of oral cancer prevention and treatment programs in northern Brazil cities. Although there are some interesting results, this study has some limitations. The small sample size, the short study interval and the paraffinization process may degenerate HPV DNA.

## 5. Conclusions

This study managed to determine the prevalence of OSCC associated with HPV infection, despite the fact that HPV prevalence had a low presence. That is a good result when we consider that HPV is an important etiological factor to oral carcinoma. Our major concern was the high prevalence of 58.5%, which presents concerning data to our region, demonstrating that the lower prevalence presented in previous studies might have some bias that directly influenced the results. So, we identified the need to improve diagnosis and therapy services for various lesions of the oral cavity in the Pará state in order to provide greater assistance to the population of Pará and prevent oral cancer.

## Figures and Tables

**Figure 1 pathogens-11-01106-f001:**
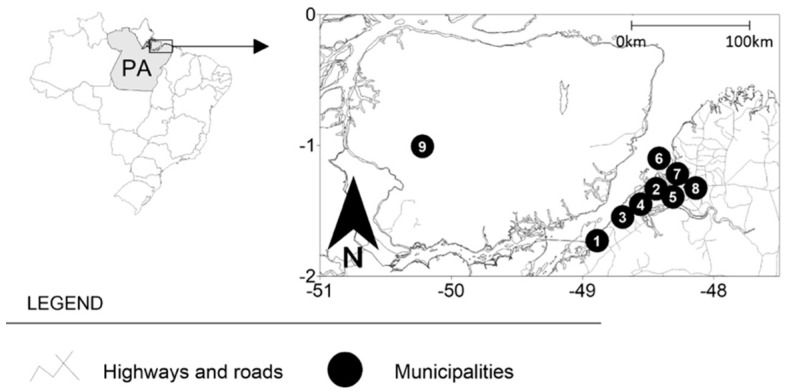
Brazil -> Pará (PA) -> Cities: (1) Abaetetuba, (2) Ananindeua, (3) Barcarena, (4) Belém, (5) Marituba, (6) Mosqueiro District, (7) Santa Bárbara, (8) Santa Izabel, and (9) Marajó Archipelago.

**Table 1 pathogens-11-01106-t001:** Sociodemographic and behavioral data.

Parameters	Total (*n* = 101)	Malignant (*n* = 59)	Benign (*n* = 42)	*p* Value
Gender				
Female	55	29 (49.2%)	26 (62%)	0.2866 ^a^
Male	46	30 (50.8%)	16 (38%)	
Ethnicity †				
White	37	22 (37.2%)	15 (35.8%)	0.9818 ^b^
Black	15	9 (15.5%)	6 (14.2%)	
Mixed	46	26 (44%)	20 (47.6%)	
Indigenous	3	2 (3.3%)	1 (2.4%)	
Age (years)				
18–29	20	5 (8.4%)	15 (35.8%)	0.0004 ^b^
30–39	9	2 (3.3%)	7 (16.6%)	
40–49	13	8 (13.5%)	5 (12%)	
50–59	16	9 (15.5%)	7 (16.6%)	
60–69	25	19 (32.2%)	6 (14.2%)	
70–79	7	6 (10.1%)	1 (2.4%)	
≥80	11	10 (17%)	1 (2.4%)	
Smoking				
Yes	71	37 (62.7%)	34 (81%)	0.2687 ^a^
No	30	22 (37.3%)	8 (19%)	
Alcohol overconsumption				
Yes	71	44 (74.5%)	27 (64.3%)	0.3710 ^a^
No	30	15 (25.5%)	15 (35.7%)	
Tobacco chewing				
Yes	18	10 (17%)	8 (19%)	0.9937 ^a^
No	83	49 (83%)	34 (81%)	
Source				
Urban cities *	75	44 (58.7%)	31 (41.3%)	0.8855 ^a^
Rural cities ‡	26	15 (57.7%)	11 (42.3)	
Belém *	36	20 (33.8%)	16 (38%)	0.9419 ^b^
Ananindeua *	20	13 (22%)	7 (16.6%)	
Marituba *	9	5 (8.4%)	4 (9.5%)	
Santa Bárbara *	10	6 (10.1%)	4 (9.5%)	
Mosqueiro ‡	4	1 (1.7%)	3 (7.1%)	
Abaetetuba ‡	6	4 (6.7%)	2 (4.7%)	
Santa Izabel ‡	9	5 (8.4%)	4 (9.5%)	
Marajó ‡	3	2 (3.3%)	1 (2.4%)	
Barcarena ‡	4	3 (5.6%)	1 (2.4%)	
Sexually active				
Yes	90	50 (84.7%)	40 (95.2%)	0.1647 ^b^
No	11	9 (15.3%)	2 (4.8%)	
Condom use				
Always	76	47 (79.6%)	29 (69%)	0.4118 ^b^
Sometimes	22	10 (17%)	12 (28.5%)	
Never	3	2 (3.4%)	1 (2.5%)	
Partners n°/past year				
1	61	41 (69.5%)	20 (%)	0.0446 ^a^
>2	40	18 (30.5%)	22 (%)	
Previous STI’s diagnosis				
Yes	49	28 (47.5%)	21 (50%)	0.9601 ^a^
No	52	31 (52.5%)	21 (50%)	

^a^ Chi-square test; ^b^ G test; * Belém, Ananindeua, Marituba and Santa Bárbara; ‡ Mosqueiro, Abaetetuba, Santa Izabel, Marajó and Barcarena. † Sociodemographic and behavioral data.

**Table 2 pathogens-11-01106-t002:** Anatomical sites of biopsy to histopathological exam.

Parameters	Total	Malignant (*n* = 59)	Benign (*n* = 42)	*p* Value
Anatomic region				
Upper gingiva	46	24 (40.6%)	22 (52.3%)	0.4713 ^b^
Lower gingiva	17	11 (18.7%)	6 (14.2%)	
Tongue	19	12 (20.3%)	7 (16.9%)	
Lips and buccal mucosa	16	10 (17%)	6 (14.2%)	
Oropharynx	3	2 (3.4%)	1 (2.4%)	

^b^ G test.

**Table 3 pathogens-11-01106-t003:** Histopathological results.

Parameters	Total	Malignant (*n* = 59)	Benign (*n* = 42)	*p* Value
Histopathological diagnosis				
Oral Squamous Cell Carcinoma	40	40 (67.7%)	-	<0.0001 ^b^
Leukoplakia	9	9 (15.2%)	-	
Erythroplakia	7	7 (11.8%)	-	
Oral Lichen planus	2	2 (3.3%)	-	
Carcinoma in situ	1	1 (2%)	-	
Traumatic fibroma	20	-	20 (47.6%)	
Verruca vulgaris	3	-	3 (7.1%)	
Focal epithelial hyperplasia	5	-	5 (12%)	
Papilloma	4	-	4 (9.5%)	
Condyloma acuminatum	8	-	8 (19%)	
Pyogenic granuloma	1	-	1 (2.4%)	
Periodontal abscess	1	-	1 (2.4%)	

^b^ G test.

**Table 4 pathogens-11-01106-t004:** HPV genotypes and results in histopathological samples.

Parameters	Malignant (*n* = 59)	Benign (*n* = 42)	*p* Value
HPV negative	42 (71.2%)	32 (76.2%)	0.6520 ^c^
HPV positive	17 (28.8%)	10 (23.8%)	
Oral Squamous Cell Carcinoma	12 (70.5%)	-	
Leukoplakia	4 (23.5%)	-	
Erythroplakia	1 (6%)	-	
Papilloma	-	3 (30%)	
Verruca vulgaris	-	3 (30%)	
Condyloma Acuminatum	-	4 (40%)	
Genotypes			
High oncogenic risk (*n* = 17)			
HPV-16	10 (58.8%)		
HPV-18	5 (29.4%)	-	
HPV-52	1 (5.9%)	-	
HPV-58	1 (5.9%)	-	
Low oncogenic risk (*n* = 10)			
HPV-6	-	6 (60.0%)	
HPV-11	-	3 (30.0%)	
HPV-42	-	1 (10.0%)	

^c^ Fisher Exact Test.

## Data Availability

All data referred to this study are available in the manuscript.

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
