# Peer review of "Prevalence and Genotyping of HPV in Oral Squamous Cell Carcinoma in Northern Brazil"

_pathogens, 2022, doi:10.3390/pathogens11101106_

Round 1
Reviewer 1 Report (New Reviewer)
This study reports the low prevalence of HPV in oral lesions in a small north Brazilian clinical cohort. Similar to other previous studies in South America, they found 29% of the 59 precancers/cancers and 10 (67%) benign lesions tested HPV-positive. No correlation between HPV-positivity and oral cancers could be determine probably due to low cohort numbers. These data add to the literature on HPV prevalence and the role of alcohol consumption in oral cancers in Brazil.
HPV genotyping has been carried out and is mentioned in the text but should also be shown in a Table.
The Introduction contains several errors in understanding human papillomavirus. For example, line 57, E6 and E7 are viral oncoproteins but L1 and L2 are not; line 87, E2 is expressed in an acute, transient viral infection; line 89, HPV proteins are not detected in serum.
The paper is written in a confusing manner e.g. lines 72-74 do not make sense. This may be due to use of English language.
Author Response
Dear Editor and Reviewers
Pathogens
Please find enclosed the revised manuscript (Manuscript ID: pathogens-1860312) entitled “Prevalence and genotyping of HPV in oral squamous cell carcinoma in Northern Brazil”.
We were pleased to see that the reviewers suggested some revisions of the manuscript. All corrections made are highlighted in the manuscript and the authors thanks the reviewer for the interesting suggestions, which have improved the manuscript.
We hope the manuscript will be suitable for publication in Pathogens – section: Emerging Pathogens – Special Issue: Emerging and Re-emerging Viral Infectious Diseases.
After reading both reviewers similar comments we authors highlighted in green all the changes we made in the paper to demonstrate the accepted reviewers suggestions to improve the manuscript and increase paper acceptance.
Hoping to hear from you at your earliest convenience, we thank you for your kind attention.
Reviewer 2 Report (New Reviewer)
1. The introduction is very poorly written, especially about HPV. This part needs to completely rewritten.
2. The citations can be improved. Some statements are not entirely right and lacks proper citations, especially in the introduction part (lines 55 - 84). Please make sure to include right citations.
3. In some places, the sentences are way too long and difficult to understand. Lacks punctuation.

Author Response
Dear Editor and Reviewers
Pathogens
Please find enclosed the revised manuscript (Manuscript ID: pathogens-1860312) entitled “Prevalence and genotyping of HPV in oral squamous cell carcinoma in Northern Brazil”.
We were pleased to see that the reviewers suggested some revisions of the manuscript. All corrections made are highlighted in the manuscript and the authors thanks the reviewer for the interesting suggestions, which have improved the manuscript.
We have hope the manuscript will be suitable for publication in Pathogens – section: Emerging Pathogens – Special Issue: Emerging and Re-emerging Viral Infectious Diseases.
After reading both reviewers similar comments we authors highlighted in green all the changes we made in the paper to demonstrate the accepted reviewers suggestions to improve the manuscript and increase paper acceptance.
Hoping to hear from you at your earliest convenience, we thank you for your kind attention.
Round 2
Reviewer 2 Report (New Reviewer)
The authors have made required changes as highlighted by my previous comments. I do not see any major concerns
This manuscript is a resubmission of an earlier submission. The following is a list of the peer review reports and author responses from that submission.
Round 1
Reviewer 1 Report
The manuscript describes the prevalence of benign and malignant oral lesions in patients treated at an oral pathology department at a dental college in Belém in Northern Brazil. Moreover, the authors assessed the prevalence of HPV in these lesions. The study has flaws in the design and statistics and lacks information regarding methodology. The manuscript and its results in the present form seem more to be oriented to oral medicine than virology.
Introduction
1) The introduction is too long describing HPV and should be more focused on the aims of the study.
2) The term alcoholism is a psychiatric disease. I would recommend using “alcohol overconsumption” or similar terms.
Methodology:
3) The authors state “This descriptive, cross-sectional, interventional study was population-based on clinical symptoms, sociodemographic and epidemiological data from individuals attended in oral pathology department at a dental college located in the city of Belém, Pará, northern Brazil”. However, the study is not an interventional study and population-based is neither an appropriate term. Cross-sectional single-centre study would be a better to use for the study.
4) How the cohort was selected is difficult to read out from the methodology. The authors state “The sample consisted of patients registered and treated at the oral pathology department in Belém, Pará” and “The study was carried out with 101 individuals who sought the aforementioned oral pathology department” How was the patient cohort selected? E.g. consecutive patients during a certain time period? Is this a referral centre or do patients book an appointment themselves?
5) What was the reason for referral to the clinic?
6) Line 109: Reference 20 seems not correct.
7) It is difficult to understand the following sentence “The oral tumors were divided by its type of oral lesion and are presented as 1- benign lesions…. 2-malignant lesions”. A lesion can be, e.g., a tumor, ulcer or infection. It is better to change the sentence to “lesions were sub-divided into benign lesions, e.g.,… and malignant lesions, e.g.,…”.
8) It is stated in table 3- “Histopathological results”, however, it is difficult to understand how lesions were diagnosed. Can the authors describe in detail how lesions were diagnosed (macroscopically, microscopically. with cultures (candida) PCR (herpes))?
9) Were also benign lesions biopsied?
10) What HPV types were tested?
11) What was the rationale for using G-test, Fisher´s test and chi-square tests for the different tables?
12) What does the exclusion criterion “individuals who were transferred to other locations” mean?
13) Were all patients who signed the informed consent biopsied? If not, age could be a bias selecting the ones to be biopsied.
Results
14) Table 1: The table template heading has not been removed.
15) Table 1: The percentages for “number of partners” are missing.
16) Table 1. The authors show a map over Brazil and the different cities. The cities seem to be on a 100 km stretch. The authors compare the fraction of benign vs. malign lesions between the different cities and there is no statistical difference. Is there a difference in the constitution of the population? If countryside cities are alike and urban cities alike, I suggest to subdivide into urban rural cities since the number of cases is low.
17) Table 1. “History of STI´s”-how was this question/these questions asked?
18) Table 2. It is stated “jaw” and ”mandible”. Do the authors mean “maxilla (upper jaw)” and mandible? The sub-divisions are not appropriate since jaw and mandible refer more to bone than mucosa. It would be better to use more conventional subdivisions (floor of mouth, tongue, gingiva etc).
19) Table 2. G-test was selected and not Fishers, why?
20) Table 3. I have difficulties understanding the statistics-according to methodology-the authors seem to have grouped the different conditions into “benign” and “malignant” and thereafter done a G-test. Statistics is redundant here.
21) Can the authors write a list of which lesions that were HPV-positive instead of grouping it as malignant and benign lesions?
22) Why were there so many invalid samples (25.6% for malignant and 57.2 for benign)?
23) The English needs to be improved as well as reducing the number of subsentences within each sentence.
Discussion:
1) The discussion needs to be revised thoroughly so it focuses more on the findings of the study in relation to previous studies.
2) The authors state “The present study evaluated the prevalence of oral SCC associated with HPV infection…”. However, it was not OSCC that was evaluated but all lesions related to HPV.
3) Line 274: The authors write “The present study identified a high prevalence of oral SCC 58.5% (59/101)…”. It is difficult to judge if this is high since we do not know for what reasons patients were referred to the clinic. Also, as state above 59 are both premalignant and malignant lesions. It is better to show the prevalence of HPV in the different lesions than putting both OPMD and oral cancer together.
4) The authors need to discuss why there were so many invalid samples.
5) The authors state on line 252-253 “This study also used data from the Department of Informatics of the Unified Health System (DATASUS) to establish the prevalence of oral cancer in Brazil”. It is difficult to understand what it is referred to (own findings or findings of ref 24).
Minor comments
Line 120: “Have” should be “have”
Line 295: Please change to “stated” in “The authors also statement that tobacco and ethanol…”
Line 301: Please change “thigh” to “high”
Reviewer 2 Report
The authors reported the rate of HPV in oral lesions in a specific location of the northeast of Brazil. Although the lack of data about the real prevalence of HPV infection in oral lesions in different areas of Brazil, the paper failure in some critical points.
1. It is not possible to report "prevalence" with a convenience sample. There is a clear selection bias, once the patients are treated at a specific specialized center. Moreover, the number of recruited patients is low.
2. Another important point is that the authors mixed pre-malignant lesions with oral SCC as a "malignant" group what is not acceptable. The authors also do not specify what "malignant" lesions were positive for HPV (it was really the SCC or, for example, the lichen planus)? This point is extremely important in order to better understand the results. Moreover, what is the rational for the p-value at Table 3?
3. Another relevant point is that, in low income countries as Brazil, the rate of HPV in oropharynx SCC is very low (see 10.1016/j.bjorl.2015.04.001) and it is well stablished the role of HPV in this tumor carcinogenesis. Based on that, in oral cavity this rate could be even lower and the authors could be facing another bias due to oral HPV colonization, for example.
4. The authors included oropharynx and lip samples, as well as benign lesions given the nomination of HPV prevalence in oral SCC at the title of the study. Furthermore, the number of "invalid" sample were high (~39%) what is another potential information bias.
It is important to point out that none of these potential limitations were explored by the authors at the discussion.
Based on that, I could not recommend the publication of this manuscript at Viruses, and my opinion is not based on the low rate of HPV in the population, what I believe is extremely important for the country to better locally understand the disease.